# Modality-Balanced Learning for Multimedia Recommendation

## Submission Number: 1280

## ABSTRACT

Multimedia content is of predominance in the modern Web era. Many recommender models have been proposed to investigate how to incorporate multimodal content information into traditional collaborative filtering framework effectively. The use of multimodal information is expected to provide more comprehensive information and lead to superior performance. However, the integration of multiple modalities often encounters the modal imbalance problem: since the information in different modalities is unbalanced, optimizing the same objective across all modalities leads to the under-optimization problem of the weak modalities with a slower convergence rate or lower performance. Even worse, we find that in multimodal recommendation models, all modalities suffer from the problem of insufficient optimization. To address these issues, we propose a Counterfactual Knowledge Distillation (CKD) method which could solve the imbalance problem and make the best use of all modalities. Through modality-specific knowledge distillation, CKD could guide the multimodal model to learn modality-specific knowledge from uni-modal teachers. We also design a novel generic-and-specific distillation loss to guide the multimodal student to learn wider-and-deeper knowledge from teachers. Additionally, to adaptively recalibrate the focus of the multimodal model towards weaker modalities during training, we estimate the causal effect of each modality on the training objective using counterfactual inference techniques, through which we could determine the weak modalities, quantify the imbalance degree and re-weight the distillation loss accordingly. Our method could serve as a plug-and-play module for both late-fusion and early-fusion backbones. Extensive experiments on six backbones show that our proposed method can improve the performance by a large margin.

## CCS CONCEPTS

• **Information systems** → **Recommender systems**; **Personalization**; **Multimedia and multimodal retrieval**.

## KEYWORDS

Multimedia Recommendation, Balanced Multimodal Learning, Knowledge Distillation

## 1 INTRODUCTION

Online platforms rely heavily on recommendation systems to suggest products, services, and content based on users' preferences and behaviors. Nowadays, a large portion of Internet content is

*ACM MM, 2024, Melbourne, Australia*
© 2024 Copyright held by the owner/author(s). Publication rights licensed to ACM.
ACM ISBN 978-x-xxxx-xxxx-x/YY/MM
https://doi.org/10.1145/nnnnnnn.nnnnnnn

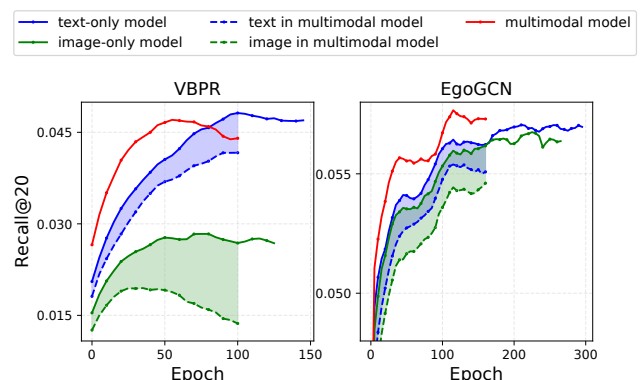

**Figure 1: A pilot study of different model variants on Amazon-Clothing. The shaded area indicates the degree of under-optimization of each modality (best viewed in color). With the use of early stopping, the training terminates at different steps, which results in the different lengths of curves.**

represented in multiple modalities, including images, texts, videos, etc. Multimedia recommendation has garnered increasing attention from researchers in recent years. Many early researches [11, 18, 31, 37, 43] follow late-fusion paradigm, which models user preferences towards each modality individually and then fuse them through summation or concatenation. Recent researchers [1, 12, 44, 52] try to adopt early-fusion which could model fine-grained inter-modal interactions and achieve better performance. For example, EgoGCN [1] performs edge-wise modulation fusion, adaptively distilling the most informative inter-modal messages to spread while preserving intra-modal processing.

The use of multimodal information is expected to provide a more comprehensive understanding of user preferences and subsequently yielding superior recommendation performance. However, recent researches [24, 32] find that optimizing the same objective for different modalities leads to the under-optimization problem of the weak modalities with slower converge rate or lower performance, which is named as modal imbalance problem. In multimedia recommendation, this phenomenon still exists and is even more pronounced since the information contained in different modalities is highly unbalanced. Taking the e-commerce scenario of the Amazon dataset [21] as an example, since the textual modality contains more detailed information such as the titles, categories, and descriptions of items, it converges faster with better performance compared with the visual modality [44]. **This gap leads to significant under-optimization issues in multimodal recommendation models.** We conduct a pilot study to validate this. Based on Figure 1 (please refer to Section 2.3 for more experiment details and findings), we could find that both visual and textual performance within the multimodal model is worse than that in the uni-modal models, i.e., visual-only and textual-only model, respectively.

To this end, recent researches [24, 32] try to modulate the learning speed of different modalities. However, these methods require an explicit distinction between the parameters of the different models and thus are only applied to late-fusion models. In this work, we propose Counterfactual Knowledge Distillation framework, CKD for brevity, to solve the imbalance problem and make the best use of all modalities for both late-fusion and early-fusion models. The overall framework of our proposed CKD is shown in Figure 2. (1) We first utilize uni-modal models as teachers to guide the multimodal student through modality-specific knowledge distillation. (2) Secondly, we design generic-and-specific distillation losses to guide the multimodal student models to learn wider-and-deeper knowledge about both universal and training triples from teachers. (3) Finally, to adaptively focus more on weaker modalities with slow converge rate, we employ counterfactual inference techniques to estimate the causal effect of each modality on the training objective, through which we could determine the weak modalities, quantify the imbalance degree and re-weight the distillation loss accordingly. Since the above operations only involve the input and output of the backbone models, CKD treats them as black-boxes, which is model-agnostic and could be served as a plug-and-play module for any existing multimedia recommendation backbones.

We conduct extensive experiments to verify the effectiveness of our proposed method on four public real-world recommendation datasets. Experimental results demonstrate that our proposed CKD gains significant improvements when plugged into six state-of-the-art models.

To summarize, the main contribution of this work is threefold:

- We argue that the existing multimodal recommendation models suffer severely from the modality imbalance problem and all modalities are under-optimized and far from reaching the upper bound of their capabilities.
- We propose a novel Counterfactual Knowledge Distillation framework which could serve as a plug-and-play module for any existing multimedia recommendation backbones to solve the imbalance problem and make the best use of all modalities.
- We perform extensive experiments on four public datasets when plugged into six backbones. The empirical results validate the effectiveness of our proposed model.

## 2 PRELIMINARIES

In this section, we first introduce the notations and formulate the multimedia recommendation task. Then, to motivate our model design, we conduct simple and intuitive experiments to show that multimodal models suffer from the imbalance problem. Finally, we analyze the imbalance problem from the optimization perspective.

### 2.1 Notations

We use the notation $\mathcal{U}$ and $\mathcal{I}$ to represent the sets of users and items, respectively. The number of items is denoted as $n = |\mathcal{I}|$. Each user $u \in \mathcal{U}$ has interacted with some items $\mathcal{I}^u$, indicating that the preference score $y_{ui} = 1$ for $i \in \mathcal{I}^u$. We represent the input ID embedding of user $u$ and item $i$ as $x_u$ and $x_i$, respectively, where $x_u, x_i \in \mathbb{R}^d$ and $d$ is the embedding dimension.

In addition to user-item interactions, each item is associated with multimodal content information. We represent the feature vector of item $i$ and the preference vector of user $u$ for modality $m$ as $e_i^m \in \mathbb{R}^{d_m}$ and $p_u^m \in \mathbb{R}^d$, where $d_m$ represents the dimension of the features and $m \in \mathcal{M}$ represents the modality. For example, the visual and textual modalities are represented by $m = v$ and $m = t$, respectively. It should be noted that our method is not limited to two modalities and can accommodate multiple modalities. The goal of multimedia recommendation approach is to predict users' preferences $\hat{y}_{ui}$ accurately by considering both user-item interactions and item multimodal content information.

### 2.2 Multimedia Recommender Framework

In this subsection, we introduce the general framework of multimedia recommender models. Our proposed method can serve as a flexible plug-and-play module for any multimedia recommender backbones. For multimedia recommender models, the calculation of the preference score of user $u$ on item $i$ can be generalized as:

$$\hat{y}_{ui} = f_\Theta(x_u, x_i, p_u^m, e_i^m), \quad m \in \mathcal{M}, \tag{1}$$

where $f_\Theta(\cdot)$ represents different methods to model user-item interactions and item multimodal content information. $\Theta$ are the trainable parameters of the models.

Multimedia recommender models adopt Bayesian Personalized Ranking (BPR) loss [27] to conduct the pair-wise ranking, which encourages the prediction of an observed entry to be higher than its unobserved counterparts:

$$\mathcal{L}_{BPR} = -\sum_{(u,i,j)\in\mathcal{D}} \ln \sigma\left(\hat{y}_{ui} - \hat{y}_{uj}\right), \tag{2}$$

where $\mathcal{D} = \{(u, i, j) | i \in \mathcal{I}_u, j \notin \mathcal{I}_u\}$ denotes the training set. $\mathcal{I}^u$ indicates the observed items associated with user $u$ and $(u, i, j)$ denotes the pairwise training triples where $i \in \mathcal{I}^u$ is the positive item and $j \notin \mathcal{I}^u$ is the negative item sampled from unobserved interactions. $\sigma(\cdot)$ is the sigmoid function.

### 2.3 Modality Imbalance Problem

We conduct a pilot study on Amazon-Clothing datasets [21] to show that both late-fusion method VBPR [11] and early-fusion method EgoGCN [1] suffer from the modality imbalance problem and are not fully optimized. We show the experimental performances of *multimodal model* and *uni-modal model (image-only and text-only models)* with the same architecture. During the training phase, for *multimodal models*, we trained it using both textual and visual input. For *uni-modal models*, we ablate the input of the other modality with the average feature vector (this keeps the other modality in-distribution, but renders them uninformative). During the inference phase, for *multimodal models*, we report both multimodal performances and uni-modal performances (dashed lines in Figure 1) that are obtained by ablating the input of other modality.

The experimental results are shown in Figure 1 (performance comparison of all backbones and datasets is shown in Table 3), and note that all methods utilize early stopping which results in the different length of curves. There are three phenomena regarding the modality imbalance problem. 1) The performance of visual and textual modality within them is worse than that of their corresponding uni-modal models, which suggests that both modalities

are under-optimized and far from reaching the upper bound of their capabilities. 2) Even worse, the text-only VBPR outperforms the multi-modal VBPR. It shows that in the multi-modal joint training process, there is a strong mutual inhibition phenomenon between the modalities, resulting in $1 + 1 < 2$. 3) Weak modals with poor performance (visual modality in the example) suffer from more serious under-optimization problems.

## 2.4 Analysis

In this subsection, we introduce the analysis of the modality imbalance problem from the optimization perspective. We show that in the multimodal joint training, the dominant modality which converges faster and has better performance would reduce the updating step and thus suppress the optimization of other modalities.

Taking VBPR [11] as an example, it simply concatenates different modalities and we can formulate the representations of users and items:

$$\bar{x}_u = x_u \parallel p_u^m, \quad \bar{x}_i = x_i \parallel \bar{e}^m, \quad m \in \mathcal{M}, \tag{3}$$

and we can re-formulate the score function as:

$$\hat{y}_{ui} = x_u^\top x_i + \sum_{m \in \mathcal{M}} p_u^{m\top} \bar{e}_i^m, \tag{4}$$

and we can also re-write the BPR loss in equation 2. For convenience, we only consider the BPR loss of one training triple $(u, i, j)$:

$$\begin{aligned}
\mathcal{L}_{uij} &= -\ln \sigma \left( \hat{y}_{ui} - \hat{y}_{uj} \right) \\
&= -\ln \sigma \left( x_u^\top x_i + \sum_{m \in \mathcal{M}} p_u^{m\top} \bar{e}_i^m - x_u^\top x_j - \sum_{m \in \mathcal{M}} p_u^{m\top} \bar{e}_j^m \right) \\
&= -\ln \sigma \left( x_u^\top x_i - x_u^\top x_j + \sum_{m \in \mathcal{M}} \left( p_u^{m\top} \bar{e}_i^m - p_u^{m\top} \bar{e}_j^m \right) \right) \\
&= -\ln \sigma \left( x_u^\top x_i - x_u^\top x_j + \sum_{m \in \mathcal{M}} S_{uij}^m \right),
\end{aligned} \tag{5}$$

we denote $S_{uij}^m = p_u^{m\top} \bar{e}_i^m - p_u^{m\top} \bar{e}_j^m = p_u^{m\top} W_m (e_i^m - e_j^m)$, where $p_u^{m\top} W_m$ are trainable parameters and can be regarded as a whole in brief. With the Gradient Descent optimization, the parameters are updated as:

$$\begin{aligned}
(p_u^{m\top} W_m)^{(t+1)} &= (p_u^{m\top} W_m)^{(t)} - \eta \frac{\partial \mathcal{L}_{uij}}{\partial S_{uij}^m} \frac{\partial S_{uij}^m}{\partial (p_u^{m\top} W_m)^{(t)}} \\
&= (p_u^{m\top} W_m)^{(t)} - \eta \frac{\partial \mathcal{L}_{uij}}{\partial S_{uij}^m} (e_i^m - e_j^m)
\end{aligned} \tag{6}$$

where $\eta$ is the learning rate. According to equation 5, the gradient $\frac{\partial \mathcal{L}_{uij}}{\partial S_{uij}^m}$ can be written as:

$$\frac{\partial \mathcal{L}_{uij}}{\partial S_{uij}^m} = \frac{1}{1 + e^{x_u^\top x_i - x_u^\top x_j + \sum_{m \in \mathcal{M}} S_{uij}^m}}. \tag{7}$$

We can find that for any two modalities $m_1$ and $m_2$, we have $\frac{\partial \mathcal{L}_{uij}}{\partial S_{uij}^{m_1}} = \frac{\partial \mathcal{L}_{uij}}{\partial S_{uij}^{m_2}}$ and thus this term forms a "brigde" between the optimization of different modalities. We can infer that if one modality $m_1$ works particularly well with superior performance, it will result in a large $S_{uij}^{m_1}$, that is, a large denominator in Equation 7, thereby

resulting in a smaller $\frac{\partial \mathcal{L}_{uij}}{\partial S_{uij}^m}$. According to equation 6, the other modalities could obtain a smaller updating step. As a result, when the training of the multimodal model is about to converge, the other modality could still suffer from under-optimized representation and need further training.

## 3 THE PROPOSED METHOD

Motivated by the above analysis, in this section, we present our proposed method CKD to alleviate the imbalance problem and make each modality fully optimized. The overall framework is shown in Figure 2. Firstly, we introduce our modality-specific knowledge distillation framework to minimize the gap between uni-modal teachers and multi-modal students. We design a generic-to-specific distillation loss to help the student model learn deeper-and-wider knowledge from teachers. Finally, inspired by the counterfactual inference techniques, we propose to quantify the degree of imbalance by estimating the causal effect of each modality, through which we can adaptively re-weight the distillation losses and thus guide the student model to focus more on the weak modalities with the under-optimization problem.

## 3.1 Modality-specific Distillation

CKD proposes to utilize uni-modal models as teachers to guide the multimodal student to avoid the imbalance problem during training. Due to the different modalities inputted, the uni-modal and multi-modal models may significantly differ in their outputs. To ease the gap between teachers and students, we propose modality-specific knowledge distillation by dividing the multimodal model into uni-modal channels and only transferring modality-specific knowledge from the corresponding teacher model.

*3.1.1 Uni-modal Teachers.* We first introduce how we train uni-modal teacher models. A direct solution is constructing specific uni-modal models. For example, we can use the most powerful uni-modal models as teachers which will undoubtedly lead to a very powerful multimodal student model. However, it is unfair for the performance comparison. In this paper, we aim to solve the imbalance problem, so we only consider training uni-modal teachers with the same model architecture as the multimodal student models, without introducing additional powerful models.

To make CKD model-agnostic which could serve as a play-and-plug module for any existing multimedia recommendation backbones, we treat backbone models as black-boxes and train uni-modal teachers by only ablating the input. Specifically, when training a teacher of modality $m_1$, we ablate the input of other modalities $\mathcal{M} - \{m_1\}$ by replacing it with its average feature vector, which could keep these modalities in-distribution, but renders them uninformative. Assuming that the backbone multimodal is represented by $f(\cdot)$, we can formulate the prediction of uni-modal teacher of modality $m_1$ according to equation 1 as:

$$\hat{t}_{ui}^{m_1} = f_{\Theta_{m_1}}(x_u, x_i, p_u^{m_1}, e_i^{m_1}, \bar{p}^m, \bar{e}^m), m \in \mathcal{M} - \{m_1\}, \tag{8}$$

where we ablate the input of other modalities by with the average feature vector: $\bar{p}^m = \frac{1}{|\mathcal{U}|} \sum_{u \in \mathcal{U}} p_u^m$ and $\bar{e}^m = \frac{1}{|\mathcal{I}|} \sum_{i \in \mathcal{I}} e_i^m$. We also adopt BPR loss in equation 2 to conduct pair-wise ranking for

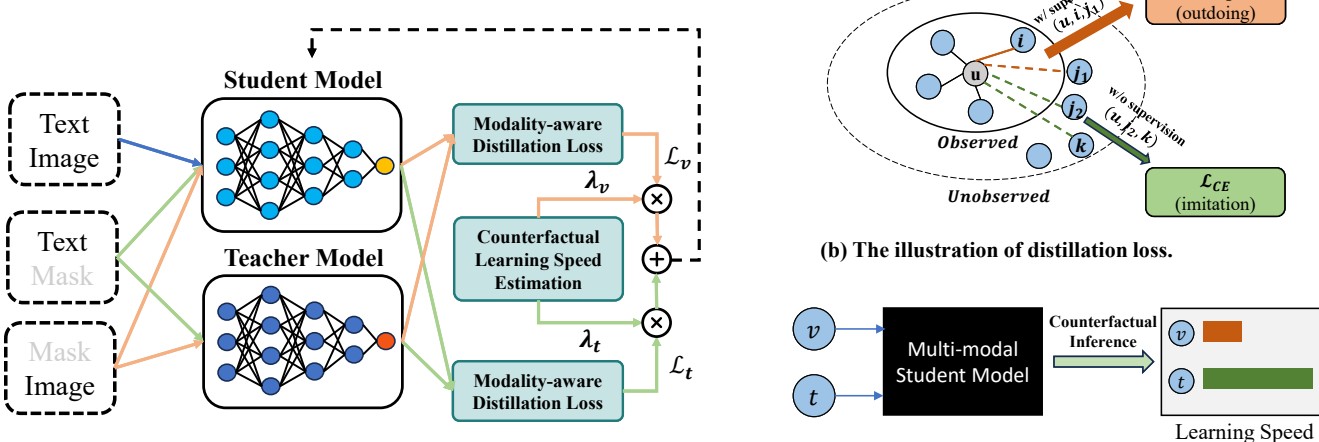

**Figure 2: (a)** An illustration of CKD model architecture. Through modality-specific knowledge distillation, CKD could guide the multimodal model to learn modality-specific knowledge from uni-modal teachers and thus alleviate competition between modalities(§3.1). **(b)** On the training triples, hinge distillation loss encourages the student model to perform better than teachers(§3.1.3) while on universal triples without supervision, CE distillation loss encourages the student model to imitate teachers (§3.1.4). **(c)** Through counterfactual inference, we can estimate the learning speed of each modality within the black-box multimodal models (§3.2).

uni-modal teachers:

$$\min_{\Theta_{m_1}} - \sum_{(u,i,j)\in\mathcal{D}} \ln \sigma\left(\hat{t}_{ui}^{m_1} - \hat{t}_{uj}^{m_1}\right). \qquad (9)$$

Similarly, we can obtain the uni-modal teacher $\Theta_m$ for each modality.

*3.1.2 Multi-modal Student.* Since we aim to only transfer modality-specific knowledge from uni-modal teachers to the multi-modal student, we also ablate the input of the multi-modal model to obtain the modality-specific prediction:

$$\hat{s}_{ui}^{m_1} = f_\Theta(x_u, x_i, p_u^{m_1}, e_i^{m_1}, \bar{p}^m, \bar{e}^m), m \in \mathcal{M} - \{m_1\}. \qquad (10)$$

We then employ a prediction-level distillation paradigm which provides a more compact representation of the teacher model's knowledge, allowing for efficient transfer without requiring the entire feature space. We design two distillation losses: (1) specific distillation loss to transfer the specific knowledge conveyed by **the triples in the training set**, and (2) generic distillation loss to transfer the deeper dark knowledge conveyed by **the universal triples** and achieve better generalization.

*3.1.3 Specific Distillation Loss.* The triples in training set $(u, i, j)$ exhibit explicit supervision signals that positive pair score $\hat{y}_{ui}$ should be larger than negative pair score $\hat{y}_{uj}$. To utilize the supervision signals, firstly, instead of directly comparing the point-wise prediction $\hat{y}_{ui}$, we compare the pair-wise ranking result $\Delta t_{uij}^m = \hat{t}_{ui}^m - \hat{t}_{uj}^m$ and $\Delta s_{uij}^m = \hat{s}_{ui}^m - \hat{s}_{uj}^m$, which could capture the relative user preferences. Furthermore, we hope that the student model not only imitates the teacher model but could surpass the teacher model by modeling the interaction between modalities. To this end, instead of

utilizing the common-used KL-divergence or Mean Squared Error loss function, we employ hinge loss to guide the student to make more informed predictions on the training triples. Hinge loss encourages the student model to perform better than the uni-model teacher on the training triple $(u, i, j)$, i.e, $\Delta s_{uij}^m > \Delta t_{uij}^m$, instead of just imitating it, i.e, $\Delta s_{uij}^m = \Delta t_{uij}^m$. Specifically, for modality $m_1$, the specific distillation loss can be formulated as:

$$\mathcal{L}_{SD}^m = \sum_{(u,i,j)\in\mathcal{D}} \max\left(\Delta t_{uij}^m - \Delta s_{uij}^m, 0\right), \qquad (11)$$

where $\mathcal{D} = \{(u, i, j)|i \in \mathcal{I}_u, j \notin \mathcal{I}_u\}$ denotes the triples in the training set. $(u, i, j)$ denotes the pairwise training triples where $i \in \mathcal{I}^u$ is the positive item and $j \notin \mathcal{I}^u$ is the negative item sampled from unobserved interactions.

*3.1.4 Generic Distillation Loss.* In addition to the explicit knowledge implied by training triples, CKD also proposes generic distillation loss to learn the wider knowledge from teacher models and achieve better generalization. Unlike specific distillation which pairs each user with one positive item and one negative item, generic distillation aims to transfer knowledge about $(u, j, k)$ where $v_j$ and $v_k$ are uniformly sampled from the item set and not paired with the user $u$. Since $v_j$ and $v_k$ are not fixed to be positive and negative, $(u, j, k)$ does not convey supervision signals and thus we cannot hope student model to perform better than teachers following equation 11. Therefore, following [40], we optimize the cross entropy-based distillation loss to align them:

$$\mathcal{L}_{GD}^m = - \sum_{(u,j,k)} \left(\Delta \bar{t}_{ujk}^m \cdot \log(\Delta \bar{s}_{ujk}^m) + (1 - \Delta \bar{t}_{ujk}^m) \cdot \log(1 - \Delta \bar{s}_{ujk}^m)\right), \qquad (12)$$

where $\Delta \bar{t}_{ujk}^m$ and $\Delta \bar{s}_{ujk}^m$ are ranking results processed by sigmoid function $\sigma(\cdot)$ with temperature $\tau$:

$$\Delta \bar{t}_{ujk}^m = \sigma(\Delta t_{ujk}^m/\tau), \quad \Delta \bar{s}_{ujk}^m = \sigma(\Delta s_{ujk}^m/\tau), \quad (13)$$

the overall distillation loss of modality $m$ can be formulated:

$$\mathcal{L}^m = \lambda_g \mathcal{L}_{GD}^m + \mathcal{L}_{SD}^m, \quad (14)$$

where $\lambda_g$ is the hyper-parameter that balances the two distillation losses. Through specific-to-generic distillation, the multi-modal student can learn both deeper-and-wider knowledge from uni-modal teachers.

## 3.2 Counterfactual Learning Speed Estimation

In this subsection, we introduce counterfactual conditional learning speed to estimate the causal effect of each modality on the joint-training objective, through which we could quantify the imbalance degree and adaptively re-weight the distillation loss of different modalities and thus guide the student model to focus more on the weak modalities with the under-optimization problem.

Since the multimodal model is trained jointly and thus the parameters are learned based on multimodal inputs, we cannot directly use the trained parameters to infer the model's casual behavior under a specific modality, especially for early-fusion methods. The counterfactual inference [28] is to answer the counterfactual question based on factual observations. For example, when we estimate the contribution of visual modality, we want to answer the question "How much improvement can the introduction of visual modality bring when other modalities remain unchanged?"

From a causal perspective, we can define the *outcome*, denoted as $Z$, as the prediction of pairwise rankings $\Delta s_{uij}$. The *treatment*, labeled as $T$, pertains to the incorporation of visual modality input. Specifically, we assign $T = 0$ to signify the absence of visual modality input and $T = 1$ to denote its inclusion. $Z(T = 1)$ represents the hypothetical outcome that would be observed if all modalities were incorporated. Conversely, $Z(T = 0)$ represents the hypothetical outcome if all other modalities were held constant while excluding the visual modality input. By contrasting the potential outcomes $Z(T = 1)$ and $Z(T = 0)$, we can assess the causal impact of integrating visual modality input on the learning objective. This comparative analysis enables us to comprehend the treatment's influence and ascertain whether the inclusion of visual modality input yields any notable changes in the predicted outcome. The *individual treatment effect* (ITE) $\delta_{(u,i,j)}^m$ of triple $(u, i, j)$ can be formulated:

$$\delta_{(u,i,j)}^m = Z(T = 1) - Z(T = 0)$$
$$= \Delta s_{(u,i,j)} - \Delta s_{(u,i,j)}^{\bar{m}}, \quad (15)$$

where $\Delta s_{(u,i,j)}^{\bar{m}}$ denotes the ranking prediction where the input modality $m$ is masked with its average feature vector. We can obtain the *Average Treatment Effect* (ATE) as the casual contribution by taking an average over the ITEs:

$$\gamma^m = \mathbb{E}_{(u,i,j) \in \mathcal{B}} \big[ \delta_{(u,i,j)}^m \big]$$
$$= \frac{1}{|\mathcal{B}|} \sum_{(u,i,j)} \delta_{(u,i,j)}^m, \quad (16)$$

**Table 1: Statistics of the datasets**

| Dataset | #Users | #Items | #Interactions | Density |
|---|---|---|---|---|
| Clothing | 39,387 | 23,033 | 237,488 | 0.00026 |
| Sports | 35,598 | 18,357 | 256,308 | 0.00039 |
| Beauty | 22,363 | 12,101 | 172,188 | 0.00064 |
| Baby | 19,445 | 7,050 | 139,110 | 0.00101 |

where $\mathcal{B}$ denotes the triple set in batch. When we consider the modality imbalance problem, another factor is that modalities convey different types of information, and the capability upper bound usually varies from different modalities. Therefore, we cannot simply compare ATEs to decide on insufficiently optimized modalities. To this end, we modify the ATE with the approximate performance upper bound, i.e. the performance of the uni-modal teacher model, to represent the relative degree of under-optimization of each modality:

$$\rho^m = \frac{\gamma^m}{\sum_{(u,i,j) \in \mathcal{B}} t_{(u,i,j)}^m}, \quad (17)$$

where a lower $\rho^m$ indicates that the modality $m$ is under-optimized and suffers more from the imbalance problem. We propose to re-weight the distillation loss of different modalities according to:

$$\lambda_{m_1} = 1 - \frac{\rho^{m_1}}{\sum_{m \in \mathcal{M}} \rho^m}, \quad (18)$$

where we emphasize the distillation loss of modalities with lower $\rho$. In this way, the overall optimization objective is presented:

$$\mathcal{L} = \mathcal{L}_{BPR} + \lambda_{kd} \sum_{m \in \mathcal{M}} \lambda_m \mathcal{L}^m. \quad (19)$$

where $\lambda_{kd}$ is the hyper-parameter that balances the learning rate between BPR loss and distillation losses.

## 4 EXPERIMENTS

In this section, we conduct experiments on four widely used real-world datasets. We first describe the experimental settings, including datasets, baselines, evaluation, and implementation details. Then, we report and discuss the experimental results to answer the following research questions:

- **RQ1:** How does CKD perform when plugged into different multimedia recommendation methods?
- **RQ2:** Does CKD achieve balanced multimodal learning?
- **RQ3:** How do different components impact CKD 's performance

Some other experiments such as hyper-parameter sensitive analysis are shown in supplementary material.

## 4.1 Experiments Settings

*4.1.1 Datasets.* We conduct experiments on four categories of widely used Amazon review dataset introduced by McAuley et al. [21]: 'Beauty', 'Baby', 'Clothing, Shoes and Jewelry', 'Sports and Outdoors', which are named as **Beauty**, **Baby**, **Clothing** and **Sports** in brief. We use the 5-core version of Amazon datasets where each user and item have 5 interactions at least. The statistics of these

**Table 2: Performance comparison of CKD when plugged into different backbones. The best performance is highlighted in bold. $\Delta Improv.$ indicates relative improvements over backbones in percentage. All improvements are significant with $p$-value $\leq 0.05$.**

| Model | Baby | | | Sports | | | Clothing | | | Beauty | | |
|---|---|---|---|---|---|---|---|---|---|---|---|---|
| | R@20 | NDCG@20 | P@20 | R@20 | NDCG@20 | P@20 | R@20 | NDCG@20 | P@20 | R@20 | NDCG@20 | P@20 |
| VBPR | 0.0462 | 0.0208 | 0.0025 | 0.0493 | 0.0219 | 0.0026 | 0.0492 | 0.0221 | 0.0025 | 0.0977 | 0.0474 | 0.0055 |
| +GB | 0.0482 | 0.0022 | 0.0026 | 0.0564 | 0.0253 | 0.0030 | 0.0551 | 0.0241 | 0.0028 | 0.0105 | 0.0523 | 0.0061 |
| +OGM | 0.0448 | 0.0199 | 0.0024 | 0.0513 | 0.0221 | 0.0028 | 0.0533 | 0.0231 | 0.0026 | 0.0962 | 0.0464 | 0.0055 |
| +CKD | **0.0568** | **0.0261** | **0.0031** | **0.0621** | **0.0285** | **0.0033** | **0.0604** | **0.0260** | **0.0029** | **0.1169** | **0.0572** | **0.0065** |
| $\Delta Improv.$ | 23.03% | 25.29% | 24.00% | 26.02% | 30.27% | 26.92% | 22.70% | 17.65% | 16.00% | 19.61% | 20.65% | 18.18% |
| DeepStyle | 0.0425 | 0.0190 | 0.0023 | 0.0471 | 0.0211 | 0.0025 | 0.0450 | 0.0192 | 0.0023 | 0.0891 | 0.0418 | 0.0050 |
| +GB | 0.0439 | 0.0191 | 0.0023 | 0.0581 | 0.0262 | 0.0031 | 0.0504 | 0.0222 | 0.0026 | 0.0101 | 0.0478 | 0.0055 |
| +OGM | 0.0445 | 0.0194 | 0.0024 | 0.0483 | 0.0214 | 0.0026 | 0.0474 | 0.0021 | 0.0025 | 0.0913 | 0.0435 | 0.0051 |
| +CKD | **0.0613** | **0.0279** | **0.0033** | **0.0684** | **0.0309** | **0.0037** | **0.0655** | **0.0290** | **0.0035** | **0.1214** | **0.0579** | **0.0065** |
| $\Delta Improv.$ | 44.24% | 47.03% | 43.48% | 45.24% | 46.37% | 48.00% | 45.60% | 50.99% | 51.78% | 36.25% | 38.54% | 30.46% |
| MMGCN | 0.0585 | 0.0247 | 0.0031 | 0.0483 | 0.0213 | 0.0026 | 0.0272 | 0.0113 | 0.0014 | 0.0704 | 0.0316 | 0.0036 |
| +GB | 0.0594 | 0.0251 | 0.0031 | 0.0493 | 0.0217 | 0.0027 | 0.0283 | 0.0119 | 0.0015 | 0.0724 | 0.0323 | 0.0037 |
| +OGM | 0.0601 | 0.0255 | 0.0033 | 0.0488 | 0.0215 | 0.0027 | 0.0293 | 0.0126 | 0.0015 | 0.0719 | 0.0330 | 0.0042 |
| +CKD | **0.0643** | **0.0279** | **0.0035** | **0.0578** | **0.0255** | **0.0032** | **0.0345** | **0.0146** | **0.0017** | **0.0820** | **0.0368** | **0.0042** |
| $\Delta Improv.$ | 9.93% | 12.86% | 13.29% | 19.67% | 19.84% | 23.93% | 26.91% | 29.27% | 21.43% | 16.46% | 16.57% | 16.98% |
| GRCN | 0.0790 | 0.0359 | 0.0042 | 0.0834 | 0.0378 | 0.0044 | 0.0637 | 0.0280 | 0.0032 | 0.1321 | 0.0647 | 0.0075 |
| +GB | 0.0794 | 0.0364 | 0.0042 | 0.0855 | 0.0387 | 0.0045 | 0.0632 | 0.0275 | 0.0032 | 0.1303 | 0.0640 | 0.0074 |
| +OGM | 0.0807 | 0.0366 | 0.0043 | 0.0851 | 0.0388 | 0.0044 | 0.0634 | 0.0276 | 0.0032 | 0.1317 | 0.0639 | 0.0075 |
| +CKD | **0.0866** | **0.0389** | **0.0047** | **0.0922** | **0.0424** | **0.0048** | **0.0667** | **0.0288** | **0.0034** | **0.1355** | **0.0659** | **0.0077** |
| $\Delta Improv.$ | 9.66% | 8.42% | 12.20% | 10.50% | 12.24% | 9.32% | 4.77% | 3.00% | 6.66% | 2.57% | 1.84% | 2.28% |
| EgoGCN | 0.0808 | 0.0366 | 0.0043 | 0.0969 | 0.0456 | 0.0051 | 0.0597 | 0.0270 | 0.0030 | 0.1391 | 0.0693 | 0.0080 |
| +GB | 0.0799 | 0.0362 | 0.0042 | 0.0968 | 0.0454 | 0.0050 | 0.0621 | 0.0283 | 0.0032 | 0.1390 | 0.0690 | 0.0080 |
| +OGM | 0.0811 | 0.0368 | 0.0043 | 0.0971 | 0.0454 | 0.0052 | 0.0600 | 0.0261 | 0.0030 | 0.1384 | 0.0689 | 0.0079 |
| +CKD | **0.0856** | **0.0375** | **0.0045** | **0.1004** | **0.0463** | **0.0053** | **0.0651** | **0.0289** | **0.0033** | **0.1440** | **0.0704** | **0.0083** |
| $\Delta Improv.$ | 5.89% | 2.47% | 4.93% | 3.61% | 1.55% | 4.35% | 9.06% | 7.21% | 9.27% | 3.52% | 1.54% | 3.87% |
| MGCN | 0.0638 | 0.0292 | 0.0034 | 0.0903 | 0.0430 | 0.0048 | 0.0745 | 0.0348 | 0.0038 | 0.1306 | 0.0636 | 0.0073 |
| +GB | 0.0633 | 0.0290 | 0.0033 | 0.0883 | 0.0423 | 0.0046 | 0.0733 | 0.0344 | 0.0036 | 0.1298 | 0.0631 | 0.0072 |
| +OGM | 0.0640 | 0.0294 | 0.0034 | 0.0899 | 0.0429 | 0.0048 | 0.0739 | 0.0346 | 0.0037 | 0.1310 | 0.0638 | 0.0074 |
| +CKD | **0.0710** | **0.0327** | **0.0038** | **0.0937** | **0.0445** | **0.0049** | **0.0770** | **0.0363** | **0.0040** | **0.1349** | **0.0651** | **0.00757** |
| $\Delta Improv.$ | 11.29% | 11.99% | 12.76% | 3.72% | 3.39% | 3.33% | 3.36% | 4.40% | 5.85% | 3.29% | 2.36% | 3.70% |

datasets are summarized in Table 1. Amazon dataset includes visual modality and textual modality. The 4,096-dimensional visual features have been extracted and published[1]. We also extract sentence embeddings as textual features by concatenating the title, descriptions, categories, and brand of each item and then utilize sentence-transformers [26] to obtain 1,024-dimensional sentence embeddings.

*4.1.2 Backbone Models.* To evaluate the effectiveness of our proposed model, we plug it into several state-of-the-art recommendation models, including late-fusion methods VBPR[11], DeepStyle [18], MMGCN[37], GRCN[36] and early-fusion methods EgoGCN[1] and MGCN[42].

*4.1.3 Baseline Models.* We also conduct experiments on two state-of-the-art baseline methods that aim to solve the imbalance problem:

---

[1]http://jmcauley.ucsd.edu/data/amazon/links.html

- **GB** [32] optimizes both uni-modal and multimodal losses simultaneously, and reweights the losses according to the overfitting-to-generalization-ratio.
- **OGM** [24] alleviates the optimization imbalance problem with on-the-fly gradient modulation to adaptively control the optimization of each modality.

Both the baselines require an explicit distinction between the parameters of the different modalities and thus are only applied to late-fusion models. For early-fusion method EgoGCN and MGCN, we implement them by roughly dividing the parameters into two parts.

*4.1.4 Evaluation and Implementations.* For each dataset, we select 80% of the historical interactions of each user to constitute the training set, 10% for validation, and the remaining 10% for the test set. We adopt three widely used metrics to evaluate the performance of preference ranking: Recall@$k$, NDCG@$k$, and Precision@$k$. By default, we set $k = 20$ and report the averaged metrics for all users in the test set. We implemented our method in PyTorch and the

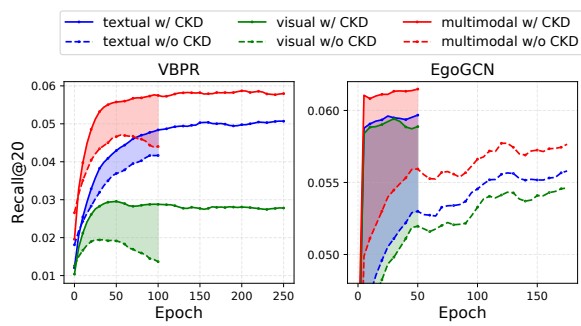

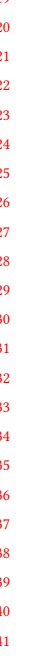

Figure 3: The performance curves during training on Amazon-Clothing dataset. The shaded area indicates the improvement of our method (best viewed in color).

embedding dimension $d$ is fixed to 64 for all models to ensure fair comparison. The optimal hyper-parameters were determined via grid search on the validation set: the learning rate is tuned amongst {0.0001, 0.0005, 0.001, 0.005}, the coefficient of $L_2$ normalization is searched in {1e-5, 1e-4, 1e-3, 1e-2}, the dropout ratio in {0.0, 0.1, $\cdots$, 0.8}, the $\lambda_g$ in {0.1, 0.5, 1, 5} and $\lambda_{kd}$ in {1e-3, 1e-2, 1e-1, 5e-1}. We set the temperature $\tau = 0.1$ in Equation 13. Besides, we stop training if recall@20 on the validation set does not increase for 10 successive epochs.

## 4.2 Overall Performance Comparison (RQ1)

We start by conducting experiments to evaluate the overall performance when baselines and our method are plugged into different backbone models. The results are reported in Table 2, from which we have the following observations:

- Our method CKD significantly outperforms baselines when plugged into different backbone models, verifying the effectiveness of our methods. Specifically, our method improves over the backbone multimedia recommendation models by 18.6%, 20.1%, 20.9%, and 15.7% in terms of Recall@20 on average on Baby, Sports, Clothing and Beauty, respectively. This indicates that our proposed method successfully mitigates the modality imbalance problem, leading to enhanced optimization of parameters for each modality in the multi-modal backbone model. Consequently, the overall recommendation performance experiences a substantial boost.
- When applied to the early-fusion backbone model EgoGCN, both GB and OGM do not yield significant improvements. In contrast, our approach not only achieves performance gains in simple late-fusion models but also demonstrates improvements in complex early-fusion models. By treating the backbone model as a black box and only applying ablating at the input end and distillation at the output end, our method is model-agnostic, offering greater flexibility.

## 4.3 Efficacy of CKD (RQ2)

*4.3.1 Uni-modal Performance Comparison.* In this subsection, we show the experimental results of each individual modality to verify the effectiveness of our method in alleviating the modality imbalance problem. We report the uni-modal performances within

Table 3: Performance comparison of individual modality of different backbones in terms of Recall@20. The best performance is highlighted in bold.

| Model | Baby | | Sports | | Clothing | | Beauty | |
|---|---|---|---|---|---|---|---|---|
| | textual | visual | textual | visual | textual | visual | textual | visual |
| VBPR | 0.0442 | 0.0352 | 0.0482 | 0.0210 | 0.0376 | 0.0202 | 0.0737 | 0.0596 |
| uni-teacher | 0.0476 | 0.0362 | 0.0543 | 0.0342 | 0.0477 | 0.0283 | 0.0955 | 0.0827 |
| GB | 0.0459 | 0.0361 | 0.0550 | 0.0256 | 0.0437 | 0.0219 | 0.0916 | 0.0707 |
| OGM | 0.0419 | 0.0362 | 0.0465 | 0.0216 | 0.0385 | 0.0216 | 0.0766 | 0.0550 |
| Ours | **0.0550** | **0.0471** | **0.0611** | **0.0445** | **0.0484** | **0.0319** | **0.0994** | **0.0914** |
| DeepStyle | 0.0221 | 0.0293 | 0.0327 | 0.0190 | 0.0283 | 0.0195 | 0.0500 | 0.0478 |
| uni-teacher | 0.0420 | 0.0332 | 0.0459 | 0.0318 | 0.0514 | 0.0304 | 0.0856 | 0.0774 |
| GB | 0.0267 | 0.0301 | 0.0434 | 0.0338 | 0.0382 | 0.0307 | 0.0614 | 0.0668 |
| OGM | 0.0233 | 0.0305 | 0.0327 | 0.0179 | 0.0299 | 0.0199 | 0.0513 | 0.0498 |
| Ours | **0.0593** | **0.0449** | **0.0519** | **0.0406** | **0.0622** | **0.0429** | **0.1031** | **0.0971** |
| MMGCN | 0.0474 | 0.0511 | 0.0514 | 0.0263 | 0.0231 | 0.0156 | 0.0580 | 0.0382 |
| uni-teacher | 0.0481 | 0.0377 | 0.0454 | 0.0339 | 0.0146 | 0.0218 | 0.0616 | 0.0569 |
| GB | 0.0548 | 0.0521 | 0.0511 | 0.0431 | 0.0226 | 0.0178 | 0.0655 | 0.0620 |
| OGM | 0.0510 | 0.0531 | 0.0433 | 0.0307 | 0.0239 | 0.0164 | 0.0594 | 0.0445 |
| Ours | **0.0569** | **0.0529** | 0.0476 | **0.0367** | **0.0311** | **0.0236** | **0.0770** | **0.0624** |
| GRCN | 0.0740 | 0.0614 | 0.0772 | 0.0613 | 0.0490 | 0.0416 | 0.1182 | 0.1039 |
| uni-teacher | 0.0770 | 0.0718 | 0.0803 | 0.0746 | 0.0547 | 0.0448 | 0.1031 | 0.1167 |
| GB | 0.0755 | 0.0739 | 0.0796 | 0.0784 | 0.0534 | 0.0488 | 0.1257 | 0.1202 |
| OGM | 0.0728 | 0.0611 | 0.0779 | 0.0646 | 0.0497 | 0.0418 | 0.1150 | 0.1011 |
| Ours | **0.0781** | **0.0760** | **0.0819** | **0.0794** | **0.0551** | **0.0498** | **0.1262** | **0.1215** |
| EgoGCN | 0.0787 | 0.0796 | 0.0949 | 0.0945 | 0.0579 | 0.0571 | 0.1370 | 0.1372 |
| uni-teacher | 0.0794 | 0.0801 | 0.0958 | 0.0967 | 0.0589 | 0.0591 | 0.1377 | 0.1370 |
| GB | 0.0772 | 0.0761 | 0.0964 | 0.0956 | 0.0595 | 0.0586 | 0.1389 | 0.1374 |
| OGM | 0.0801 | 0.0803 | 0.0953 | 0.0941 | 0.0571 | 0.0566 | 0.1385 | 0.1374 |
| Ours | **0.0841** | **0.0840** | **0.0970** | **0.0970** | **0.0608** | **0.0600** | **0.1393** | **0.1383** |
| MGCN | 0.0622 | 0.0625 | 0.0881 | 0.0895 | 0.0734 | 0.0722 | 0.1289 | 0.1292 |
| uni-teacher | 0.0631 | 0.0633 | 0.0900 | 0.0911 | 0.0742 | 0.0739 | 0.1300 | 0.1303 |
| GB | 0.0615 | 0.0618 | 0.0871 | 0.0883 | 0.0731 | 0.0719 | 0.1286 | 0.1288 |
| OGM | 0.0633 | 0.0640 | 0.0890 | 0.0904 | 0.0733 | 0.0720 | 0.1293 | 0.1299 |
| Ours | **0.0708** | **0.0704** | **0.0922** | **0.0931** | **0.0760** | **0.0755** | **0.1340** | **0.1344** |

each multimodal backbone model, the performances of uni-modal teachers, and the uni-modal performances within baselines and our proposed method. The experimental results are shown in Table 3, from which we have the following observations:

- As the phenomenon in Figure 1, the uni-modal channels within multimodal backbones exhibit sub-optimal performance in comparison to their respective uni-modal teachers. This observation suggests that despite the intended capability of multimodal models to capture complementary information across modalities, they fall short of fully harnessing the potential inherent in each modality. Consequently, both the overall performance and the performance of individual uni-modal channels fail to attain their maximum potential.
- While baseline methods have shown some enhancement in uni-modal performance, they still fall significantly short of reaching their maximum potential. This is evident from the substantial performance disparity between baselines and uni-modal teacher models.
- The uni-modal channels within CKD significantly outperform those within the vanilla backbone models, especially for the modalities that are optimized insufficiently in joint multimodal training. For example, when plugged into VBPR, our method improves 28.7% and 64.3% for textual modality and visual modality, respectively. Through knowledge distillation, we can make up for the under-optimization of

the visual modality in joint multimodal training, thereby greatly improving the information utilization rate of the visual modality and the overall recommendation performance.

- Our proposed CKD also outperforms the uni-modal teachers in most cases. Through generic and specific distillation losses, our knowledge distillation framework does not simply imitate uni-modal teachers, but improves the information utilization rate of each modality under their guidance, captures complementary information between modalities and thus can obtain recommendation performance beyond them.

*4.3.2 Performance Curves During Training.* We additionally illustrate the performance curves during the training process, as depicted in Figure 3, akin to the representation in Figure 1. It is discernible that both the overall multimodal performance and unimodal performances of our proposed methods surpass that of the backbone models. This observation serves to substantiate the efficacy of CKD.

## 4.4 Ablation Study (RQ3)

To further analyze CKD, in this subsection, we conduct component analyses by repeatedly assessing and comparing the models after removing each component. Specifically, we design the following variants of CKD:

- **w/o Gen.** removes the generic distillation loss and only retains the specific distillation loss.
- **w/o re-weight** removes the counterfactual conditional learning speed loss re-weight.
- **repl. KL-divergence/MSE Loss** replaces the hinge loss with KL-diverge or MSE loss in the specific distillation module.

Due to limited space, we only present the results of VBPR[11], MMGCN[37] and EgoGCN[1] in Table 4 and leave others in supplementary material. We can observe that each of these key components contributes substantially. The generic distillation considers more general triplets in addition to traditional triplets with positive and negative items, which greatly facilitates the deeper dark knowledge distillation. Additionally, the counterfactual conditional learning speed can monitor the discrepancy of each unimodal's contribution to the overall learning objective, which could reflect the degree of imbalance in the training process. By dynamically increasing the distillation weight of weak modalities, the optimization process of each modality is controlled and the imbalance problem can be alleviated. Finally, compared to KL-divergence and MSE losses, our proposed hinge loss could encourage the student to make more informed predictions.

## 5 RELATED WORK

### 5.1 Multimedia Recommendation

Recommendation systems have achieved significant success through the implementation of Collaborative Filtering (CF) methods. However, they face challenges in handling sparse data characterized by limited user-item interactions, as their predictive models traditionally depend on substantial interactions to generate accurate recommendations based on user preferences. To overcome this obstacle, recent researchers have incorporated diverse types of auxiliary information, such as visual images, textual descriptions, and

**Table 4: Ablation study results in terms of Recall@20. The best performance is highlighted in bold.**

| Model | Baby | Sports | Clothing | Beauty |
|---|---|---|---|---|
| VBPR + CKD | **0.0568** | **0.0621** | **0.0604** | **0.1169** |
| w/o. Gen. | 0.0549 | 0.0580 | 0.0550 | 0.1084 |
| w/o. re-weight | 0.0541 | 0.0594 | 0.0591 | 0.1120 |
| repl. KL-divergence Loss | 0.0475 | 0.0517 | 0.0520 | 0.1059 |
| repl. MSE Loss | 0.0487 | 0.0526 | 0.0592 | 0.1053 |
| MMGCN + CKD | **0.0643** | **0.0578** | **0.0345** | **0.0820** |
| w/o. Gen. | 0.0566 | 0.0493 | 0.0297 | 0.0731 |
| w/o. re-weight | 0.0605 | 0.0551 | 0.0322 | 0.0783 |
| repl. KL-divergence Loss | 0.0621 | 0.0573 | 0.0330 | 0.0747 |
| repl. MSE Loss | 0.0632 | 0.0564 | 0.0340 | 0.0796 |
| EgoGCN + CKD | **0.0856** | **0.1004** | **0.0651** | **0.1440** |
| w/o. Gen. | 0.0836 | 0.0972 | 0.0630 | 0.1422 |
| w/o. re-weight | 0.0833 | 0.0981 | 0.0622 | 0.1403 |
| repl. KL-divergence Loss | 0.0841 | 0.0915 | 0.0651 | 0.1362 |
| repl. MSE Loss | 0.0834 | 0.0857 | 0.0633 | 0.1312 |

videos [5, 6, 38]. This integration has given rise to multimedia recommendation systems, leveraging extensive multimedia content details associated with items [2–4, 8–11, 13–20, 22, 23, 25, 29, 33–37, 41, 43–51, 53]. Traditional research in multimedia recommendation [2, 3, 11, 13, 18, 47] extends the basic CF framework by incorporating multimodal content alongside item representations. More recently, there is a growing interest in GNN-based multimedia recommendation systems [15, 19, 20, 36, 37, 43] and integrating contrastive learning [10, 25, 33, 35, 41, 44, 45].

### 5.2 Balanced Multimodal Learning

Multimodal models, though expected to outperform single-modal ones by incorporating more information, have shown counter-intuitive results in previous experiments, sometimes performing even worse [24, 30, 32]. The heterogeneity of modalities poses a challenge for unified training strategies to fully exploit their potential. Recent approaches aim to address this by balancing optimization across modalities [7, 24, 30, 32, 39]. Wang et al. (2020) proposed adaptively mixing gradients from different modalities to mitigate overfitting [32]. Du et al. (2021) introduced well-trained single-modal teachers to guide outputs, recognizing sub-optimal performance in single-modal networks after joint training [7]. Peng et al. (2022) addressed the tendency of models to favor dominant modalities by balancing learning rates during training [24].

## 6 CONCLUSION

In this paper, we have proposed CKD which aims to solve the modal imbalance problem and make the best use of all modalities, which could easily serve as a plug-and-play module for any existing multimedia recommendation models. Extensive experiments on four real-world datasets and six state-of-the-art multimedia recommendation backbones have been conducted to demonstrate that CKD achieves superior performance.

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
