# OpenReview forum: "Modality-Balanced Learning for Multimedia Recommendation"
_acmmm.org/ACMMM/2024/Conference — MM2024 Oral_

### Official Review · Reviewer_7445 · 2024-05-24

**Rating:** 4
**Confidence:** 3

**Summary:**

This paper proposes a counterfactual knowledge distillation method, termed CKD, to address the issue of imbalanced modalities in multimedia recommendation. Specifically, this method distills uni-modal knowledge from teachers to the student and leverages counterfactual inference to identify weak modalities for distillation loss re-weighting.

**Strengths:**

1\. The problem of imbalanced modalities is well elucidated, supported by a pilot study.

2\. The method is well-structured, facilitating ease of reading.

3\. The overall performance improvement is significant.

**Limitations:**

1\. The pilot analysis in Section 2.4 is insufficient and misleading. Despite the same gradient for $S^m_{uij}$, the gradient for the trainable parameters differs due to the different values of $e^m_i - e^m_j$. Moreover, only the VBPR method is considered.

2\. The different loss formats for generic distillation loss are unclear. The core difference between specific and generic distillation loss is the sample methods of triples. It is unclear why a different cross-entropy loss is used for the generic loss while a hinge loss is used for the specific one. Can the generic distillation loss also use the hinge loss format? This inconsistency raises concerns.

3\. The motivation for transforming $\gamma^m$ to $\rho^m$ is unclear.

4\. The performance report in Table 3 is incorrect. OGM achieves 0.0531 in the visual Baby category, while "Ours" achieves 0.0529 but is indicated as the best performance.

5\. The suggestion to replace the generic distillation loss with a hinge loss format should be addressed, along with an analysis explaining why the generic distillation loss aims for imitation rather than outdoing, unlike the specific distillation loss.

6\. Some important works are missing, such as CR[1], EliMRec[2], and Modality Balancing[3].

[1]Wang, Wenjie, et al. "Clicks can be cheating: Counterfactual recommendation for mitigating clickbait issue." *Proceedings of the 44th International ACM SIGIR Conference on Research and Development in Information Retrieval*. 2021.

[2]Liu, Xiaohao, et al. "Elimrec: Eliminating single-modal bias in multimedia recommendation." *Proceedings of the 30th ACM International Conference on Multimedia*. 2022.

[3]Shang, Yu, et al. "Enhancing Adversarial Robustness of Multi-modal Recommendation via Modality Balancing." *Proceedings of the 31st ACM International Conference on Multimedia*. 2023.

**Suitability:**

3

---

### Official Review · Reviewer_QEGm · 2024-05-24

**Rating:** 4
**Confidence:** 2

**Summary:**

The paper proposes Counterfactual Knowledge Distillation framework, which could serve as a plug-and-play module for existing multimedia recommendation backbones and consists of three main components: (1) utilizes uni-modal models as teachers to guide the multi-modal student through modality-specific knowledge distillation. (2) designs generic-and-specific distillation losses to guide the multi-modal student models to learn wider-and-deeper knowledge about both universal and training triples from teachers. (3) employs counterfactual inference techniques to estimate the causal effect of each modality on the training objective.

**Strengths:**

(1)The paper provides a thorough analysis of the existing modality imbalance problem.
(2)The paper introduces a plug-and-play module for existing multimedia recommendation backbones to solve the imbalance problem.
(3)The paper conducts rigorous experiments to prove the validity of the proposed framework.

**Limitations:**

I think the paper is well-considered.

**Suitability:**

3

---

### Official Review · Reviewer_SaJy · 2024-05-25

**Rating:** 5
**Confidence:** 3

**Summary:**

The paper introduces a novel framework named Counterfactual Knowledge Distillation (CKD) that addresses the modal imbalance problem in multimedia recommendation systems. CKD leverages uni-modal models as teachers to guide a multimodal student model through a knowledge distillation process, enhancing the optimization of underperforming modalities. By employing counterfactual inference techniques, CKD also estimates the causal effect of each modality, enabling an adaptive focus on weaker modalities and significantly improving recommendation performance across various datasets and backbone models. The proposed method serves as an effective plug-and-play module to balance multimodal learning and boost the overall recommendation quality.

**Strengths:**

1. The motivation is clear and the idea is easy to follow;
2. This paper is written clearly, and the purpose of each section is evident;
3. This paper conducted a rich set of experiments and provided the main baselines. Moreover, the method provided have shown significant improvement compared to the baselines；

**Limitations:**

1. Figure 1 in the supplementary materials demonstrates that the choice of hyperparameters significantly impacts the results, and the optimal sets of hyperparameters vary across different methods, which to some extent limits the practicality of the approaches;
2. I would suggest providing a comparison of hyperparameter tuning for a same backbone (e.g., EgoGCN) across different datasets;

**Suitability:**

3

---

### Official Review · Reviewer_gEWx · 2024-05-25

**Rating:** 5
**Confidence:** 3

**Summary:**

This paper introduces the Counterfactual Knowledge Distillation(CKD) method, addressing insufficient optimization of multimodal representation in multimedia recommendation. It incorporates modality-specific knowledge distillation and causal inference into multimodal representation learning. Extensive experiments across four datasets showcase the effectiveness of CKD in multimedia recommendation.

**Strengths:**

1. This paper employs modality-specific knowledge distillation to enhance multimodal representation. Moreover, it introduces counterfactual inference to adaptively re-weight the distillation loss across different modalities.
2. The paper is well-organized.
3. The evaluation utilizes various datasets and models.

**Limitations:**

1. As shown in section 2.4, does parallel multimodal representation learning also lead to this problem?

2. More baselines need to be included such as BM3[1], SLMRec[2], and so on.

> [1] Xin Zhou, Hongyu Zhou, Yong Liu, Zhiwei Zeng, Chunyan Miao, Pengwei Wang, Yuan You, and Feijun Jiang. 2023. Bootstrap latent representations for multi-modal recommendation. In Proceedings of the ACM Web Conference 2023. 845–854
[2] Zhulin Tao, Xiaohao Liu, Yewei Xia, Xiang Wang, Lifang Yang, Xianglin Huang, and Tat-SengChua. 2023. Self-Supervised Learning for Multimedia Recommendation.  IEEE Trans. Multim. 25(2023), 5107–5116.

3. More detailed analysis in Section 4.2 is needed. For instance, why does the performance improvement vary across different models?

4. The conclusions presented in this paper are insufficient to provide comprehensive and effective information.

5. The authors should clarify why only VBPR and EgoGCN are utilized in Section 4.3.2. Are the performances of other baselines comparable?

**Suitability:**

3

---

### Meta-Review · Area_Chair_L1g8 · 2024-06-30

**Recommendation:** Accept (Oral)
**Confidence:** 5

**Metareview:**

This paper proposed a novel Counterfactual Knowledge Distillation (CKD) method to address the modal imbalance problem. Based on the reviewers’ comments, the motivation is clear, and the paper is well-organized. However, there are still areas that need improvement. I recommend that the authors revise their paper according to the reviewers’ comments. At this time, I lean towards acceptance.